# Monocytes Count, NLR, MLR and PLR in Canine Inflammatory Bowel Disease

**DOI:** 10.3390/ani14060837

**Published:** 2024-03-08

**Authors:** Maria Chiara Marchesi, Giulia Maggi, Valentina Cremonini, Arianna Miglio, Barbara Contiero, Carlo Guglielmini, Maria Teresa Antognoni

**Affiliations:** 1Department of Veterinary Medicine, University of Perugia, Via San Costanzo 4, 06126 Perugia, Italy; giulia.maggi@studenti.unipg.it (G.M.); cremonini.valentina2@gmail.com (V.C.); maria.antognoni@unipg.it (M.T.A.); 2Department of Animal Medicine, Production and Health, University of Padua, Viale dell’Università 16, 35020 Padua, Italy; barbara.contiero@unipd.it (B.C.); carlo.guglielmini@unipd.it (C.G.)

**Keywords:** chronic enteropathy, IBD, gastroenterology, hematological ratios, monocytes, dog

## Abstract

**Simple Summary:**

Inflammatory bowel disease (IBD) is a common chronic gastrointestinal disorder in dogs. Hematological ratios, in particular red blood cell distribution width (RDW), neutrophil-to-lymphocyte ratio (NLR), monocyte-to-lymphocyte ratio (MLR), and platelet-to-lymphocyte ratio (PLR), are considered cost-effective predictors of clinical disease activity in human patients with IBD and Crohn’s disease. This study evaluated the diagnostic value of hematological parameters and ratios in 85 client-owned dogs, comparing 60 dogs with IBD to 25 clinically healthy dogs. Monocytes count and the hematological ratios MLR, NLR, and PLR can be useful in the diagnostic work-up of dogs with IBD.

**Abstract:**

This is an observational retrospective study on 85 client-owned dogs, 60 with IBD and 25 clinically healthy dogs. This study aims to assess the clinical relevance of some easy to obtain and cost-effective hematological parameters including red blood cell distribution width (RDW), neutrophil-to-lymphocyte ratio (NLR), monocyte-to-lymphocyte ratio (MLR), and platelet-to-lymphocyte ratio (PLR) in dogs with IBD. Comparison of clinical and laboratory parameters between dogs with IBD and control dogs was carried out and the ability to distinguish between these two groups of dogs was evaluated by calculating the area under the receiver-operating characteristic curve (AUCROC). Univariate and multivariable logistic regression analysis estimated the odds ratio (OR) of developing IBD with a 95% confidence interval (CI). MLR and monocytes count had the highest accuracy in facilitating the discrimination of dogs with IBD from control dogs with an AUCROC of 0.839 and 0.787 at the cut-off of >0.14% and >3.7 cells*10^2^/µL, respectively. According to two multivariable models, monocytes count (OR = 1.29; *p* = 0.016), NLR (OR = 1.80; *p* = 0.016), and MLR > 0.14 (OR = 8.07; *p* < 0.001) and PLR > 131.6 (OR = 4.35; *p* = 0.024) were significant and independent predictors of IBD for models one and two, respectively. Monocytes count and the hematological ratios MLR, NLR, and PLR can be useful in the diagnostic work-up of dogs with IBD.

## 1. Introduction

Chronic enteropathies (CEs) encompass a group of disorders marked by persistent or recurrent gastrointestinal (GI) signs lasting at least 3 weeks, associated with intestinal inflammation but lacking clear evidence of an identifiable cause [1]. From a clinical perspective, canine CEs can be classified into three main categories: food-responsive (FRE), antibiotic-responsive (ARE), and immunosuppressant-responsive enteropathy (IRE) [2]. Dogs that do not respond to specific treatments fall under the classification of non-responsive enteropathy (NRE) [3]. When protein loss across the intestinal wall results in hypoalbuminemia and hypoglobulinemia, the condition is termed protein-losing enteropathy (PLE) [1]. Data regarding the true prevalence of CE in dogs are limited [3]. Epidemiological studies have reported a prevalence range of 0.35–2.9% for canine CEs [4,5,6].

Inflammatory bowel disease (IBD) encompasses a cluster of idiopathic, chronic gastrointestinal disorders characterized by mucosal inflammation and a favorable response to immunosuppressive agents [7]. Consequently, in veterinary medicine, the terms IBD and IRE can be used interchangeably [2,3]. Diagnosis typically relies on exclusion, necessitating a comprehensive diagnostic work-up to eliminate other known causes of chronic gastrointestinal signs [1]. The exact cause of IBD remains elusive. The gathering of data and evidence in companion animals suggests that this intestinal inflammation stems from an altered interplay between environmental factors, the intestinal microbiota, and the mucosal immune system in a susceptible host, implicating genetic factors [1,7,8].

Various clinical, laboratory, and histopathologic indices have been proposed to characterize disease activity in dogs with IBD [9,10,11,12,13,14]. Notably, the canine IBD activity index (CIBDAI) and canine chronic enteropathy activity index (CCECAI) are widely used to predict the severity and outcome in dogs affected by CEs [9,10]. Moreover, several laboratory parameters, including serum albumin (Alb), cobalamin, folate, C-reactive protein, canine pancreatic lipase immunoreactivity, and fecal calprotectin have been assessed as predictive and prognostic markers in dogs with CEs [9,10,11,12,13,14].

Recently, studies have explored the utility of hematological parameters and ratios as novel predictive and prognostic markers in various pathological conditions, both in humans and dogs [2,15,16,17,18,19,20,21]. Specifically, red blood cell distribution width (RDW), neutrophil-to-lymphocyte ratio (NLR), monocyte-to-lymphocyte ratio (MLR), platelet-to-neutrophil-ratio (PNR), and platelet-to-lymphocyte ratio (PLR) are considered cost-effective predictors of clinical disease activity in human patients with IBD and Crohn’s disease [20,21,22,23,24,25]. Similarly, in dogs, a preliminary study has suggested that NLR serves as an inflammatory marker, offering insights into disease severity in cases of IBD [2]. The aim of this study is to assess the clinical significance of different hematological parameters and ratios in dogs with IBD. We hypothesize that these simple and cost-effective tests may play a valuable role in distinguishing between clinically healthy dogs and those with IBD.

## 2. Materials and Methods

### 2.1. Animals

This retrospective observational study was conducted at a single veterinary teaching hospital (VTH) from January 2016 to June 2023. A cohort of dogs diagnosed with IBD (IBD group) was assembled. The inclusion criteria comprised: (1) a definitive diagnosis of IBD based on a chronic history (lasting at least 3 weeks) of GI signs, including anorexia, reduced appetite, vomiting, diarrhea, and weight loss; (2) exclusion of FRE; (3) exclusion of ARE; (4) negative results from diagnostic tests eliminating other disorders linked with chronic gastrointestinal signs, such as mechanical gastrointestinal obstruction, gastrointestinal neoplasia, exocrine pancreas insufficiency, pancreatitis, pancreatic tumors, hepatic disease, and hypoadrenocorticism; and (5) confirmation of intestinal inflammation through histological findings.

For each dog, the diagnostic procedures included a complete blood cell (CBC) and biochemical profile, including serum trypsin-like-immunoreactivity (TLI), serum cobalamin, serum folate, and basal cortisol, along with an abdominal ultrasound. A dietetic trial, involving hydrolyzed food for a minimum of 2 weeks, was conducted to exclude FRE. An antibiotic trial, employing metronidazole at 10 mg/kg q12h for 2 weeks, was performed to rule out ARE. Dogs exhibiting exocrine pancreatic insufficiency, hypoadrenocorticism, or a positive response to either the dietary or antibiotic trial were excluded from the study. In cases where IBD was suspected, confirmation was sought through histological examination of endoscopic GI biopsies, including the gastric and duodenal regions. Additional biopsies from the ileum and colon were obtained if deemed necessary based on clinical examination. Notably, none of the dogs in this group had received immunosuppressant treatment in the 3 weeks leading up to endoscopy. A histological examination was conducted by a single pathologist following the guidelines outlined by the World Small Animal Veterinary Association (WSAVA) [7].

A control group of clinically healthy dogs was assembled from animals participating in the VTH’s blood donor program. Criteria outlined in the Italy Ministry of Health Guidelines were followed for the selection of donor dogs. Specifically, these dogs were aged between 2 and 8 years, had a body weight exceeding 25 kg, and were regularly vaccinated. Each donor animal underwent a thorough assessment for clinical health, involving a comprehensive clinical history, physical examination, and a complete CBC, biochemical profile, and urinalysis. Additionally, donor dogs were confirmed to be free from blood-borne diseases, as evidenced by a negative serological test for *Leishmania infantum*, *Anaplasma phagocytophilum*, *Ehrlichia canis*, *Dirofilaria immitis*, and *Babesia* spp.

### 2.2. Data Collection

Data for the IBD group were extracted from the electronic internal medical database encompassing information such as age, sex, breed, weight, CIBDAI [9], CCECAI [10], WSAVA histopathological grading [7], CBC results, and serum total protein (TP) and Alb. These data for the IBD group were recorded at the time of endoscopy (T0). The data for the control group, also retrieved from the electronic internal medical database, related to the screening visit required for inclusion in the group of donors for the hospital’s blood bank. This information included age, sex, breed, weight, CBC results, and serum TP and Alb.

### 2.3. Laboratory Findings

CBC and biochemical profiles were conducted using standard procedures at the diagnostic laboratory of the VTH. Hematological variables including red blood cell (RBC), RDW, hematocrit (Hct), hemoglobin (Hgb), mean corpuscular volume (MCV), mean corpuscular hemoglobin concentration (MCHC), white blood cell (WBC), neutrophils, lymphocytes, platelets, and mean platelets volume (MPV) were measured utilizing a laser hematology analyzer (Sysmex XT-2000iV; Sysmex, Kobe, Japan), previously validated for canine hematology [26]. The analyzer was equipped with multispecies software, and quality control and calibration were performed weekly using e-check Xe (Sydmex).

The NLR was calculated as the absolute value of neutrophils divided by the absolute value of lymphocytes. The MLR was calculated as the ratio between the absolute value of monocyte and lymphocyte values. The PLR was determined as the absolute value of platelets divided by the absolute value of lymphocytes. The PNR was calculated as the absolute value of platelets divided by the absolute value of neutrophils.

Biochemical parameters, including TP and Alb, were measured using automated chemistry analyzers (Hitachi 904, Boehringer Mannheim and Seac Radim reagents, Biolabo sas, Les Hautes, France).

### 2.4. Statistical Analysis

Statistical analyses were conducted using commercially available software packages (SAS 9.4 Copyright © 2002–2012 by SAS Institute Inc., Cary, NC, USA; MedCalc Statistical Software version 16.4.3, MedCalc Software, Ostend, Belgium). Continuous data were assessed for distribution using a Shapiro-Wilk test. Results are presented as either the mean and standard deviation or the median and interquartile range for normally or non-normally distributed data, respectively. Categorical variables are reported as the number and percentage within each category.

Data extracted from case records included information on sex, breed, age, and body weight (BW). For breed, dogs were classified into two categories: purebred and crossbred. Laboratory variables encompassed RBC, Hct, Hgb, MCV, MCHC, RDW, WBC, neutrophils, lymphocytes, monocytes, eosinophils, platelets, NLR, MLR, PNR, PLR, as well as serum TP and Alb levels. A comparison of clinical and laboratory variables between dogs with IBD and control dogs was conducted using the Student’s t-test, the Mann-Whitney non-parametric test, or the zeta test for two proportions, depending on whether the variables were continuous (normally or not normally distributed) or categorical. The Spearman rank test was employed to assess the association between laboratory variables and the CIBDAI and CCECAI scores in dogs with IBD.

To evaluate the discriminatory ability of key continuous hematological variables that exhibited significant differences between the two groups in the univariate descriptive analysis (namely, RBC, Hgb, WBC, Neutrophils, Monocytes, NLR, MLR, PNR, and PLR), we utilized receiver operating characteristic (ROC) curve analysis. The area under the curve (AUCROC) was calculated, along with the corresponding binomial exact confidence interval (CI). Specifically, sensitivity and specificity were computed at various cut-off points. The Youden index was used to identify cut-off points that maximized sensitivity and specificity.

Univariate and multivariable logistic regression analyses were used to estimate the odds ratio (OR), along with a 95% CI for each variable’s association with the risk of diagnosing IBD. Continuous variables that showed significant association with IBD in the univariate analysis (*p* < 0.05) were included in a multivariable model. The stepwise backward elimination method was applied after considering autocorrelation and interaction between predictors. Furthermore, we examined dichotomous variables derived from the ROC curve analysis: BW (≤21.9 kg vs. >21.9 kg), RBC (≤7.13 cells*10^6^/µL vs. >7.13 cells*10^6^/µL), Hgb (≤16.9 g/dL vs. >16.9 g/dL), WBC (>9.58 cells*10^3^/µL vs. ≤9.58 cells*10^3^/µL), neutrophils (>8.56 cells*10^3^/µL vs. ≤8.56 cells*10^3^/µL), monocytes (>3.7 cells*10^2^/µL vs. ≤3.7 cells*10^2^/µL), NLR (>4.18 vs. ≤4.18), MLR (>0.14 vs. ≤0.14), and PLR (>131.6 vs. ≤131.6). Subsequently, we constructed the multivariable model using dichotomous variables significantly associated with IBD in the univariate analysis.

For all analyses, a significance level of *p* < 0.05 was applied.

## 3. Results

### 3.1. Study Population and Laboratory Parameters

The study involved a total of 85 dogs, including 60 dogs diagnosed with IBD and 25 clinically healthy dogs. Of these, 35 were females and 50 were males. Of all the involved dogs, 36 were neutered; of these, 19 were female and 17 were males. The breed distribution within the IBD group was varied, with the most prevalent breeds being mixed breed (18 dogs), German Shepherd (14 dogs), and Jack Russell Terrier and Golden Retriever (4 dogs each). Additionally, several other breeds were represented by a few cases each.

The study population had an average age of 5.9 ± 3.4 years and an average BW of 22.3 ± 11.6 kg. Dogs diagnosed with IBD exhibited a median CIBDAI score of 5 (range 1–11) and a median CCECAI score of 5 (range 1–15). A summary of signalment data categorized by group is presented in Table 1. Notably, dogs with IBD had a lower BW (18.4 ± 10.3 kg) compared to control dogs (31.9 ± 8.6 kg). However, no statistically significant age difference was observed between the two groups.

Among the continuous variables, dogs with IBD displayed significant reductions in mean RBC (*p* = 0.012), Hct (*p* = 0.020), Hgb (*p* = 0.006), and lymphocytes (*p* = 0.043) when compared to control dogs. Conversely, dogs with IBD exhibited notable elevations in median WBC (*p* = 0.008), neutrophils (*p* = 0.009), monocytes (*p* < 0.001), NLR (*p* = 0.001), MLR (*p* < 0.001), and PLR (*p* = 0.012) relative to control dogs (Table 1).

The Spearman rank correlation analysis revealed a weak yet statistically significant negative correlation between serum albumin levels and both CIBDAI and CCECAI scores (rho = −0.268, *p* = 0.04; rho = −0.308, *p* = 0.018, respectively). Furthermore, a negative correlation was identified between serum protein levels and CCECAI score (rho = −0.263, *p* = 0.044).

Table 2 and Appendix A present the findings from the ROC curve analyses. Notably, the MLR displayed the highest accuracy in predicting the presence of IBD, yielding an AUC of 0.839 (95% CI 0.744–0.910). Monocyte count exhibited an AUCROC of 0.787 (95% CI 0.684–0.868), with cut-off values of >0.14 and >3.7 cells*10^2^/µL, respectively.

### 3.2. Univariate and Multivariable Analysis

The univariate logistic regression analysis, which included both continuous variables and dichotomized variables, revealed significant correlations between the presence of IBD and certain hematological parameters. In the univariate analysis that included continuous variables, negative correlations were observed between the presence of IBD and BW (*p* < 0.001), RBC (*p* = 0.048), and Hgb (*p* = 0.032). Positive correlations were evident with WBC (*p* = 0.011), neutrophils (*p* = 0.009), monocytes (*p* = 0.010), NLR (*p* = 0.006), MLR (*p* = 0.002), and PLR (*p* = 0.022) (Table 3).

Similarly, for dichotomized variables, positive correlations were identified between IBD presence and RBC ≤ 7.13 cells*10^6^/µL (*p* = 0.022), Hgb ≤ 16.9 g/dL (*p* = 0.023), WBC > 9.58 cells*10^3^/µL (*p* = 0.007), monocytes >3.7 cells*10^2^/µL (*p* = 0.001), MLR > 1.44 (*p* = 0.002), and PLR > 154.9 (*p* = 0.002) (Table 4).

Building on the insights gained from the logistic univariate analyses, two robust multivariable models were constructed to identify independent predictors of IBD while addressing multicollinearity concerns (Table 5). In model one, the inclusion of continuous variables Hgb, neutrophils, monocytes, NLR, and PLR validated increased monocytes and NLR as independent risk factors for IBD presence (OR = 1.29 and 1.8, 95% CI 1.02–1.62 and 1.11–2.92; *p* = 0.031 and *p* = 0.016, respectively). In model two, the significance of MLR > 0.14 (OR = 8.07, 95% CI 2.59–25.14; *p* < 0.001) and PLR > 131.6 (OR = 4.35, 95% CI 1.21–15.67; *p* = 0.024) as robust, independent predictors of IBD was underscored.

## 4. Discussion

In this study, various hematological parameters in dogs with IBD were compared with those of clinically healthy dogs to evaluate their diagnostic value as cost-effective and easily accessible biomarkers for canine IBD. Our results indicate that monocytes count, NLR, MLR, and PLR were independently associated with canine IBD. Moreover, both MLR and monocytes count exhibited high accuracy in discriminating between dogs with IBD and clinically healthy dogs.

The substantial cohort of dogs with IBD, consisting of 60 individuals, included characteristics consistent with previous studies. The most frequently affected breeds were mixed breed dogs (30%) and German Shepherds (23.3%), aligning in part with current literature indicating an increased susceptibility to IBD in German shepherds [27,28]. Notably, a recent study identified an association in German Shepherds with IBD and the low expression of Toll-like receptor 4 and Toll-like receptor 5, which are responsible for activating the innate immune system [28]. The mean age of the IBD group (6.06 ± 3.91 years) was consistent with age ranges reported in dogs with CE [29]. The IBD group had a significantly lower BW compared to the control group. It is essential to note that control dogs, being part of the hospital’s blood donor program, must meet weight requirements for donation. This discrepancy may introduce a selection bias between the two groups.

Our results indicate that the IBD group exhibited significantly lower values for RBC, Hct, Hgb, and lymphocytes compared with control dogs. This is attributed to non-regenerative anemia, reflective of chronic inflammation or enteric blood loss, commonly observed in the course of IBD [30]. Conversely, dogs with IBD showed significantly elevated values for WBC, neutrophils, monocytes, NLR, MLR, and PLR compared to control dogs. Lymphocytopenia, monocytosis, and neutrophilia are part of the physiological response of the immune system to systemic inflammation and stress [2]. In inflammatory states, the neutrophil count may increase, while the lymphocyte count may decrease, a phenomenon termed “stress leukogram”. Simultaneously, neutrophilia, with or without a left shift, is associated with erosive/ulcerative intestinal lesions [30]. Monocytosis can occur in small animals during chronic inflammatory states, associated with necrosis and immuno-mediated disease [31]. Therefore, chronic inflammation in the course of IBD may manifest as an increase in WBC (neutrophilia and monocytosis) [2,30]. Conversely, in the course of IBD, abnormal lymphocyte function in peripheric circulation and at the intestinal mucosal level, due to the loss of CD4+ T cells and lymphangiectasia, can occur [3]. Increased PLT count is a possible hematologic manifestation observed in dogs with IBD as an expression of chronic inflammation [2,30].

Among hematological parameters that were significantly elevated in dogs with IBD, those demonstrating superior diagnostic accuracy were monocytes count, NLR, MLR, and PLR. Given the pathogenesis mechanisms mentioned above, it is evident how these hematological parameters and ratios may be influenced by ongoing IBD, although they are not specific to this disease. Benvenuti et al. (2020) were the first to establish the NLR as a prognostic indicator of canine IBD, and linked it to clinical severity measured by CCECAI, protein loss, and histological lymphangiectasia [2]. These authors supported the hypothesis regarding the potential use of NLR as a marker of disease severity in canine IBD [2]. From our results, positive correlations were observed only in the univariate analysis with continuous variables; unfortunately, we have not identified a predictive cut-off for this hematological ratio. However, our work represents a subsequent phase of analysis as we employed multivariable analysis models. In fact, the statistically significant variables identified in the univariate analysis allowed us to construct two models using multivariable analysis. In model one, which included continuous variables, monocytes and NLR were identified as independent risk factors for the presence of IBD. The results of model one support NLR and monocytosis as predictors of IBD in dogs: for each increment of one unit (corresponding to an increment of 100 cells/µL for monocytes), we observed an 80% and 29% increase, respectively, in the probability for the animal to have IBD. In model two, which included dichotomized variables, an MLR ratio >0.14 and a PLR ratio >131.6 were identified as independent predictors for canine IBD. In dogs, MLR has the potential to be used as a biomarker of severity and treatment responses, as seen in dogs with myxomatous mitral valve disease [15]. To the best of our knowledge, this is the first study showing MLR ratio as a biomarker of IBD in dogs. MLR is an inflammatory marker [15], and in the course of IBD it may increase in relation to an increase in monocytes count and a decrease in lymphocyte count. In our study, model two identified a correlation between MLR and IBD in dogs. The probability of having IBD was eight times higher for dogs with MLR > 0.14. Regarding platelet count and platelet ratio, our study also supports the use of PLR as a biomarker of canine IBD because the probability of having IBD was four times higher for dogs with PLR > 131.6. This can be correlated to an increase in platelet count in the course of inflammatory states associated with CE, and to the decrease in lymphocytic count often observed in the course of IBD. Thrombocytosis is a hematologic abnormality in dogs associated with various inflammatory conditions and immune-mediated disease [32]. In veterinary medicine, PLR was identified as a biomarker of inflammatory states associated with numerous diseases (e.g., acute pancreatitis, myxomatous mitral valve disease, neoplastic condition) in small animals [15,18,33]. Additionally, Cristóbal et al. used NLR and PLR as inflammatory biomarkers in a dog with CE treated with adipose-derived mesenchymal stem cells [34]. In their study, NLR and PLR decreased significantly over time during the treatment, indicating a reduction in the inflammatory condition associated with CE [34].

This study has several limitations. Firstly, the retrospective nature of the study compelled us to include only dogs with complete medical records, leading to the exclusion of many dogs with suspected IBD due to missing data in their medical records. From the retrospective nature of the study arises another important limitation, namely the absence of follow-up. This has prevented us from comparing hematological parameters and ratios at the time of diagnosis with those during the disease. Consequently, it hinders the evaluation of their usefulness at the prognostic level. We acknowledge the significant limitations of this study; however, we consider it a first step that can pave the way for subsequent assessments of the use of these parameters for prognostic purposes during IBD. In our comparison of different hematological parameters in dogs with IBD and clinically healthy dogs to assess diagnostic value, we did not include a comparison with other CEs such as FRE and ARE, nor other GI diseases (e.g., neoplastic diseases). Another limitation stems from the type of patient at our hospital, a VTH serving as a referral center, where many included dogs were referrals from veterinary practitioners. It is possible that a few animals had received glucocorticoid treatment, which could have influenced some hematological alterations such as leucopenia and thrombocytosis [31,32]. However, no dog that had received immunosuppressant treatment during the 3 weeks prior to endoscopy was included in the study. In our study, the presence of ARE was ruled out using an antibiotic trial and not by prebiotic or fecal microbiome transplantation (TMF), as suggested by recent studies [35,36]. TMF has not been performed in the patients being studied due to the absence of a donor who meets the requirements defined by Chaitman et al. [37]. Additionally, the use of prebiotics and TMF as a treatment for IBD patients has started to be defined over the last decade. Hence, the retrospective nature of the study and the need for a homogeneous group in the diagnostic process and therapies imposed the impossibility of including IBD dogs treated with prebiotics only. Finally, an important limitation arises from the non-homogeneity between the IBD and control groups. The control group, comprised largely of large breed dogs due to the inclusion criteria for blood donors, resulted in statistical analyses showing a correlation between the weight of the animals and the presence of IBD.

## 5. Conclusions

In conclusion, our study suggests that monocytes count and the hematological ratios NLR, MLR, and PLR might serve as useful predictors to differentiate between dogs with IBD and healthy dogs. While an increase in these parameters is not specific, it was independently associated with IBD. The main limitation of this study is the lack of specificity in the results, due to comparison with a control group consisting of only healthy dogs. As a result, this potentially diminishes the clinical significance of this study and its relevance. Indeed, the primary objective of our study was to assess the potential utility of ongoing hematological ratios in canine IBD. We are committed to conducting future prospective studies that will delve deeper into these findings. These forthcoming studies will delve into changes in these hematological parameters during treatment in dogs with IBD, shedding light on their potential prognostic role. Additionally, we aim to explore comparisons with other CEs and neoplasms of the GI tract, and the use of MTF and probiotics as an alternative to antibiotics in the exclusion of ARE.

## Figures and Tables

**Table 1 animals-14-00837-t001:** Clinical and laboratory variables in clinically healthy dogs and dogs with inflammatory bowel disease.

Variable	Control Dogs (*n* = 25)	IBD (*n* = 60)	*p* Value
Age (year)	5.66 ± 1.52	6.06 ± 3.91	0.503
Body weight (kg)	31.9 ± 8.6	18.4 ± 10.3	<0.001
Sex (F/M)	11/14	24/36	0.921
Purebred/Mixed breed	21/4	42/18	0.284
Total protein (g/L)	6.20 ± 0.64	5.78 ± 1.18	0.037
Albumin (g/L)	3.21 ± 0.37	2.98 ± 0.81	0.068
RBC (10^6^/μL)	7.18 ± 0.73	6.61 ± 1.30	0.012
Hemoglobin (g/dL)	17.21 ± 1.82	15.67 ± 3.22	0.006
Hematocrit (%)	46.69 ± 4.42	43.35 ± 8.52	0.020
MCV (fL)	65.08 ± 2.90	65.97 ± 4.41	0.278
MCHC (g/dL)	36.85 ± 1.13	36.05 ± 1.89	0.018
RDW (%)	15 (14.9–16.7)	15.2 (13.9–16.9)	0.674
WBC (10^3^/μL)	8.26 (7.4–10.8)	10.8 (7.89–14)	0.008
Neutrophils (10^3^/μL)	5.53 (4.33–7.41)	7.17 (5.13–9.83)	0.009
Lymphocytes (10^3^/μL)	2.40 ± 0.56	2.07 ± 0.88	0.043
Monocytes (10^2^/μL)	2.5 (1.6–3.1)	4.5 (3.05–8.3)	<0.001
Eosinophils (10^2^/μL)	4.7 (3.3–6.7)	3.9 (1.9–6.7)	0.466
Platelets (10^3^/μL)	255 ± 76	296 ± 130	0.074
NLR	2.44 (1.87–3.08)	3.39 (2.34–5.81)	0.001
MLR	0.1 (0.07–0.13)	0.25 (0.15–0.48)	<0.001
PNR	47.9 ± 18.2	44.8 ± 28.2	0.549
PLR	109.1 (84.2–126.3)	135.4 (95.5–232.5)	0.012

Continuous normally and non-normally distributed variables are expressed as mean and standard deviation or as median (interquartile range), respectively. F: female; M: male; RBC: red blood cell; MCV: mean corpuscular volume; MCHC: mean corpuscular hemoglobin concentration; RDW: red blood cell distribution width; WBC: white blood cell count; NLR: Neutrophil-to-Lymphocyte ratio; MLR: Monocyte-to-Lymphocyte ratio; PNR: Platelet-to-Neutrophil ratio; PLR: Platelet-to-Lymphocyte ratio.

**Table 2 animals-14-00837-t002:** Diagnostic accuracy of hematological variables to predict development of inflammatory bowel disease in 60 dogs.

Variable	AUCROC ± SE	95% CI	*p* Value	Cutoff	Sensitivity (%)	Specificity (%)
RBC (10^6^/µL)	0.638 ± 0.061	0.527–0.740	0.024	≤7.13	63 (50–75)	64 (42–82)
Hgb (g/dL)	0.645 ± 0.061	0.534–0.746	0.017	≤16.9	63 (50–75)	64 (42–82)
WBC (10^3^/µL)	0.684 ± 0.059	0.574–0.781	0.002	>9.58	65 (52–77)	68 (47–85)
Neutrophils (10^3^/µL)	0.680 ± 0.059	0.570–0.777	0.002	>8.56	35 (23–48)	100 (86–100)
Monocytes (10^2^/µL)	0.787 ± 0.054	0.684–0.868	<0.001	>3.7	42 (29–55)	100 (86–100)
NLR	0.731 ± 0.055	0.623–0.821	<0.001	>4.18	57 (43–69)	96 (77–100)
MLR	0.839 ± 0.049	0.744–0.910	<0.001	>0.14	77 (64–87)	88 (69–98)
PLR	0.675 ± 0.059	0.564–0.772	0.003	>131.6	55 (42–68)	84 (64–96)

RBC: red blood cell; Hgb: Hemoglobin; WBC: white blood cell count; NLR: Neutrophil-to-Lymphocyte ratio; MLR: Monocyte-to-Lymphocyte ratio; PLR: Platelet-to-Lymphocyte ratio; AUCROC: area under the curve; SE: standard error; CI: confidence interval.

**Table 3 animals-14-00837-t003:** Results of the univariate logistic regression analysis of continuous variables, showing the association between the risk for developing inflammatory bowel disease in 60 dogs.

Variable	Odds Ratio	95% CI	*p* Value
Age (year)	-	-	0.619
Sex (Female)	-	-	0.732
Purebred	-	-	0.187
Body weight (kg)	0.85	0.78–0.92	<0.001
RBC (10^6^/μL)	0.62	0.39–0.99	0.048
Hct (%)	-	-	0.073
Hgb (g/dL)	0.81	0.67–0.98	0.032
MCV (fL)	-	-	0.357
MCHC (g/dL)	-	-	0.057
RDW (%)	-	-	0.614
Total protein (g/dL)	-	-	0.101
Albumin (g/dL)	-	-	0.172
WBC (10^3^/μL)	1.26	1.05–1.50	0.011
Neutrophils (10^3^/μL)	1.34	1.07–1.68	0.009
Lymphocytes (10^3^/μL)	-	-	0.094
Monocytes (10^2^/μL)	1.35	1.07–1.69	0.010
Eosinophils (10^2^/μL)	-	-	0.826
Platelets (10^3^/μL)	-	-	0.149
NLR	1.97	1.22–3.19	0.006
MLR (%) *	1.10	1.04–1.17	0.002
PNR	-	-	0.609
PLR	1.01	1.01–1.02	0.022

* The MLR was expressed as a percentage because of the low value of this ratio. RBC: red blood cell; Hct: hematocrit; Hgb: Hemoglobin; MCV: mean corpuscular volume; MCHC: mean corpuscular hemoglobin concentration; RDW: red blood cell distribution width; WBC: white blood cell count; NLR: Neutrophil-to-Lymphocyte ratio; MLR: Monocyte-to-Lymphocyte ratio; PNR: Platelet-to-Neutrophil ratio; PLR: Platelet-to-Lymphocyte ratio; CI: confidence interval.

**Table 4 animals-14-00837-t004:** Results of the univariate logistic regression analysis of dichotomous variables showing the association between the risks for developing inflammatory bowel disease in 60 dogs.

Variable	Category	Odds Ratio (95% CI)	*p* Value
Body weight (kg)	≤21.9	-	0.948
	>21.9		
RBC (10^6^/µL)	≤7.13	3.07 (1.16–8.11)	0.023
	>7.13		
Hgb (g/dL)	≤16.9	3.07 (1.16–8.11)	0.023
	>16.9		
WBC (10^3^/µL)	>9.58	3.95 (1.46–10.67)	0.007
	≤9.58		
Neutrophils (10^3^/µL)	>8.56	-	0.943
	≤8.56		
Monocytes (10^2^/µL)	>3.7	31.38 (3.98–247)	0.001
	≤3.7		
NLR	>4.18	-	0.937
	≤4.18		
MLR	>0.14	10.4 (3.47–31.12)	<0.001
	≤0.14		
PLR	>131.6	6.42 (1.96–20.97)	0.002
	≤131.6		

RBC: red blood cell; Hgb: Hemoglobin; WBC: white blood cell count; NLR: Neutrophil-to-Lymphocyte ratio; NLR: Monocyte-to-Lymphocyte ratio; NLR: Platelet-to-Lymphocyte ratio; CI: confidence interval.

**Table 5 animals-14-00837-t005:** Results of the multivariable logistic regression analysis for the risk of developing inflammatory bowel disease in 60 dogs.

Predictors	Odds Ratio	95% CI	*p* Value
Model 1			
Monocytes (10^2^/µL)	1.29	1.02–1.62	0.031
NLR	1.80	1.11–2.92	0.016
Model 2			
MLR > 0.14	8.07	2.59–25.14	<0.001
PLR > 131.6	4.35	1.21–15.67	0.024

NLR: Neutrophil-to-Lymphocyte ratio; MLR: Monocyte-to-Lymphocyte ratio; PLR: Platelet-to-Lymphocyte ratio; CI: confidence interval.

## Data Availability

The original contributions presented in the study are included in the article/Appendix A; further inquiries can be directed to the corresponding author/s.

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
