# Peer review of "Monocytes Count, NLR, MLR and PLR in Canine Inflammatory Bowel Disease"

_animals, 2024, doi:10.3390/ani14060837_

Round 1
Reviewer 1 Report
Comments and Suggestions for Authors
Simple summary, abstract and keywords:
· Important keywords are missing; for example: IBD, monocytes, etc –please revise.
Introduction:
· Line 48 – the digit 3 is out of context – please revise.
Materials and methods:
· Line 88 – please replace blood with blond.
· Lines 127-128 – does the MLR ratio was also calculated based on absolute values? - Please address this issue and specify the type of values that were used.
Results:
· Table 1:
o Please replace limphocytes with lymphocytes.
o Please state which of the values presented as means and which are presented as medians.
o Please make sure that ratios are calculated using the same value type (means or medians, but not mean and median for a single ratio).
o Please also refer to the changes in biochemical values presented in table 1, and not only to the hematological values in the paragraph prior to table 1.
· Line 220 – the sentence is not clear. Please revise.
Discussion:
· Well written
Author Response
We thank the Reviewer for their insightful suggestions. We have amended the manuscript following their suggestion.
Comment:
Important keywords are missing; for example: IBD, monocytes, etc –please revise.
Response:
We have added "IBD," "hematological ratios," "monocytes," and "dog" as keywords.
Comment:
Line 48 – the digit 3 is out of context – please revise.
Response:
Number 3 is a reference.
Comment:
Line 88 – please replace blood with blond.
Response:
We have correct blood with blond.
Comment:
Lines 127-128 Does the MLR ratio was also calculated based on absolute values? Please address this issue and specify the type of values that were used.
Response:
MLR was calculated as the ratio between the absolute value of monocyte and lymphocyte values. This sentence was added to the manuscript.
Comment:
Table 1 - Please replace limphocytes with lymphocytes.
Response:
We have correct limphocytes whit lymphocytes.
Comment:
Table 1 - Please state which of the values presented as means and which are presented as medians.
Response:
Thank you for the requested specification. We have changed the first phrase of the legend of Table 1 as follows “Continuous normally and non-normally distributed variables are expressed as mean and standard deviation or as median (interquartile range), respectively.
Comment:
Table 1 - Please make sure that ratios are calculated using the same value type (means or medians, but not mean and median for a single ratio).
Response:
The values reported in Table 1 were not calculated as ratios between the mean or median values of the aggregated data. First, we calculated the ratio of each variable for each individual dog based on the individual values. In other words, we inserted each individual’s original value into the numerator and denominator of each ratio rather than a mean or median value. We then summarized these ratios and applied the appropriate descriptive statistics, mean or median, depending on whether the ratios were normally distributed or not.
Comment:
Table 1- Please also refer to the changes in biochemical values presented in table 1, and not only to the hematological values in the paragraph prior to table 1.
Response:
Biochemical parameters were available only for pathological subjects (IBD groups), making it impossible to compare with healthy subjects. A negative correlation between albumin and total protein levels has been identified with CIBDAI and CCECAI, as reported in lines 201-205.
Comment:
Line 220 – the sentence is not clear. Please revise.
Response:
The sentence has been changed as follows: “In the univariate analysis that included continuous variables, negative correlations were observed between the presence of IBD and BW (p < 0.001), RBC (p = 0.048), and Hgb (p = 0.032). Positive correlations were evident with WBC (p = 0.011), neutrophils (p = 0.009), monocytes (p = 0.010), NLR (p = 0.006), MLR (p = 0.002), and PLR (p = 0.022) (Table 3).”
Reviewer 2 Report
Comments and Suggestions for Authors
Title: “different hematologic parameters” is not a very helpful title. Different from what? Please be more specific.
General Comments:
This paper has a fundamental flaw limiting its usefulness, which is not unique to this study, but occurs in a lot of similar research. Ultimately, a test that discriminates dogs with IBD from healthy dogs is useless in clinical practice. No one is trying to determine if a dog is healthy or has IBD, because that is obvious from the history and physical exam. Rather, a control group composed of dogs with other GI disease (ie infectious, neoplastic, non-IBD chronic enteropathy, etc) is needed, because the ability to differentiate IBD from other chronic GI disease is what is actually relevant. This must be discussed as a major flaw in the limitations of this study, and some justification provided for why this study design is useful. Please add significant discussion of this limitation and why it is useful clinically to differentiate these groups.
Line 26: AUCROC is a more standard abbreviation
Line 40: This is incorrect. The definition given here is for Inflammatory Bowel Disease, not chronic enteropathy. Chronic enteropathies can have a known cause; IBD is the subtype of chronic enteropathy that does not. The reference given does not support this statement.
Line 46: Hypoalbuminemia and hypoproteinemia is a bit redundant. Perhaps you mean hypoglobulinemia?
Line 81: This implies all of these criteria were needed for a diagnosis of IBD. Is this the case, or was it only some subset?
Line 83: I would clarify that not all endocrine diseases were excluded- really, only Addison’s was
Line 92: Provide a justification for every dog receiving a trial of metronidazole rather than probiotics fecal transplantation, or other bacteriotherapy measures in your discussion
Line 96: What regions of the GI tract were biopsied?
Line 99: When was the last donation for these dogs? Was the iron status of the dogs known? Blood donors may have changes in hematologic parameters due to frequent blood collection, especially if bled recently relative to sampling. Please provide a justification for using blood donors as the control group rather than a more random population of healthy dogs.
Line 106: Were the clinical dogs also tested for these infectious issues? If not, clarify why it is relevant that the healthy dogs were negative
Line 113: This information on immunosuppression should be in the above section on animals
Line 127: You specify “absolute value” for some of these and not others. Be consistent
Line 175: Was any power analysis performed? If so, specify. If not, perform one and report if you reached a significant power
Line 176: Were any of these animals neutered?
Line 186: Significantly lower- you cannot say they are reduced unless you have a baseline to compare to
Line 206: Are you able to provide ROC curves for the most relevant values? Or for all of the values, as a supplemental file?
Line 218: I would remove “significant insights” and leave that kind of assessment for the discussion. Just report the results here
Line 220: This sentence doesn’t quite make sense, reword
Line 277: I don’t like the “likely” here; was the anemia usually non regenerative or not?
Line 289: Peripheral, not peripheric
Line 292: This line is superfluous
Line 298: This is a restatement of the previous sentence
Line 298-306: This is basically just a restatement of the results; you do not need to restate these findings, and in fact it makes this harder to read. Just discuss their significance, limitations, etc.
Line 305: With, not whit
Line 307: Pathogenic, not pathogenesis
Line 255: Can you provide any literature to suggest glucocorticoids >3w ago would affect these results?
Comments on the Quality of English Language
There are many spelling and grammar errors. I have pointed these out in many cases, but others are present. Please read through carefully and correct these.
Author Response
We thank the Reviewer for their insightful suggestions. We have amended the manuscript following their suggestion.
Comment:
Title: “different hematologic parameters” is not a very helpful title. Different from what? Please be more specific.
Response:
Title has been changed as follows: Monocytes count, NLR, MLR and PLR in Canine Inflammatory Bowel Disease.
General Comments:
This paper has a fundamental flaw limiting its usefulness, which is not unique to this study, but occurs in a lot of similar research. Ultimately, a test that discriminates dogs with IBD from healthy dogs is useless in clinical practice. No one is trying to determine if a dog is healthy or has IBD, because that is obvious from the history and physical exam. Rather, a control group composed of dogs with other GI disease (ie infectious, neoplastic, non-IBD chronic enteropathy, etc) is needed, because the ability to differentiate IBD from other chronic GI disease is what is actually relevant. This must be discussed as a major flaw in the limitations of this study, and some justification provided for why this study design is useful. Please add significant discussion of this limitation and why it is useful clinically to differentiate these groups.
Response:
Thanks for your valuable suggestions, we are aware of some important limitations of the study, which derive mostly from the retrospective nature of the same, and from the data that retrospectively we could extrapolate from computer medical records. The following sentence has been incorporated into the manuscript. “From the retrospective nature of the study arises another important limitation, namely the absence of follow-up. This has prevented us from comparing hematological parameters and ratios at the time of diagnosis with those during the disease. Consequently, it hinders the evaluation of their usefulness at the prognostic level. We acknowledge the significant limitation of this study; however, we consider it the first step that can pave the way for subsequent assessments of the use of these parameters for prognostic purposes during IBD.”
We also added other gastrointestinal diseases such as neoplastic in the sentence (line 349).
Comment:
Line 26: AUCROC is a more standard abbreviation.
Response:
The correct abbreviation has been introduced throughout the text.
Comment:
Line 40: This is incorrect. The definition given here is for Inflammatory Bowel Disease, not chronic enteropathy. Chronic enteropathies can have a known cause; IBD is the subtype of chronic enteropathy that does not. The reference given does not support this statement.
Response:
We have modified the text in line with suggestions of reviewer.
Comment:
Line 46: Hypoalbuminemia and hypoproteinemia is a bit redundant. Perhaps you mean hypoglobulinemia?
Response:
We have changed hypoproteinemia with hypoglobulinemia.
Comment:
Line 81: This implies all of these criteria were needed for a diagnosis of IBD. Is this the case, or was it only some subset?
Response:
All these criteria were necessary for the diagnosis of IBD.
Response:
Line 83: I would clarify that not all endocrine diseases were excluded- really, only Addison’s was
Response:
Endocrinopathy has been replaced with hypoadrenocorticism in the text for greater clarity.
Comment:
Line 92: Provide a justification for every dog receiving a trial of metronidazole rather than probiotics fecal transplantation, or other bacteriotherapy measures in your discussion.
Response:
We integrated the discussion and conclusions with the reasons for the non-use of fecal transplantation and probiotics.
Comment:
Line 96: What regions of the GI tract were biopsied?
Response:
Biopsies were performed in the stomach and duodenum. Based on the patient's clinical presentation, biopsies were also conducted at the level of the ileum and colon in some cases. The text has been integrated with this information.
Comment:
Line 99: When was the last donation for these dogs? Was the iron status of the dogs known? Blood donors may have changes in hematologic parameters due to frequent blood collection, especially if bled recently relative to sampling. Please provide a justification for using blood donors as the control group rather than a more random population of healthy dogs.
Response:
The hematological data related to the control group pertain to the screening visit necessary for inclusion in the group of donors for the hospital's blood bank. Consequently, these animals are never actually donated. This information has been added to the text for clarity.
Comment:
Line 106: Were the clinical dogs also tested for these infectious issues? If not, clarify why it is relevant that the healthy dogs were negative
Response:
Sick dogs were not tested for infectious diseases. The group of healthy control animals underwent testing for infectious diseases as a requirement for inclusion in the hospital's donor program.
Comment:
Line 113: This information on immunosuppression should be in the above section on animals
Response:
We moved the sentence to the animal section.
Comment:
Line 127: You specify “absolute value” for some of these and not others. Be consistent
Response:
Now we specify “absolute value” for all ratios.
Comment:
Line 175: Was any power analysis performed? If so, specify. If not, perform one and report if you reached a significant power
Response:
We did not perform a preliminary power analysis. Based on our results, the difference between the control group and dogs with IBD was statistically significant for the NLR variable. Considering a mean difference between the two groups equal to 1 and a standard deviation equal to 1, n=17 samples for each group of dogs are sufficient to guarantee a power of 80% and an error of 5% for the statistical tests used.
Comment:
Line 176: Were any of these animals neutered?
Response:
There were 36 dogs neutered, of these 19 were female and 17 were males. This information has been added in the text.
Comment:
Line 186: Significantly lower- you cannot say they are reduced unless you have a baseline to compare to
Response:
We remove significantly by the sentence.
Comment:
Line 206: Are you able to provide ROC curves for the most relevant values? Or for all of the values, as a supplemental file?
Response:
We have provided the ROC curves for the most relevant variables, namely those reported in Table 2, in a Supplemental file.
Comment:
Line 218: I would remove “significant insights” and leave that kind of assessment for the discussion. Just report the results here
Response:
We have changed the sentence.
Comment:
Line 220: This sentence doesn’t quite make sense, reword
Response:
The sentence has been reworded.
Comment:
Line 277: I don’t like the “likely” here; was the anemia usually non regenerative or not?
Response:
We have removed 'likely'. There was a non-regenerative anemia.
Comment:
Line 289: Peripheral, not peripheric
Response:
We have corrected the grammar errors.
Comment:
Line 292: This line is superfluous
Response:
We have removed the sentences.
Comment:
Line 298: This is a restatement of the previous sentence
Response:
We have removed the sentence.
Comment:
Line 298-306: This is basically just a restatement of the results; you do not need to restate these findings, and in fact it makes this harder to read. Just discuss their significance, limitations, etc.
Line 305: With, not whit
Line 307: Pathogenic, not pathogenesis
Response:
We have removed this section leaving out the following part inherent in the discussion of the significance of these parameters during IBD.
Comment:
Line 255: Can you provide any literature to suggest glucocorticoids >3w ago would affect these results?
Response:
I cannot provide a specific reference to the need to stop glucocorticoid therapy for 3 weeks. We considered it essential for patients to be free of therapy during this period to avoid potential misdiagnosis, especially considering the effects of glucocorticoid therapy on hematological changes (e.g., neutrophilia) and the challenge in distinguishing between IBD and lymphoma.
Round 2
Reviewer 1 Report
Comments and Suggestions for Authors
The subject of this manuscript is interesting and important.
Currently, this manuscript major drawback (to which the authors are aware and addressed in the discussion and conclusion sections), is the lack of the results' specificity, due to comparison to control group consists of only healthy dogs (no other chronic enteropahty etiologies such as: FRE and ARE, were compared). As a result, this lowers significantly the clinical significance of this study and its relevance (the knowledge obtained in this manuscript does not change the current diagnosis, treatment plan and prognosis of CE patients).
Moreover, the manuscript is not in the scope of this special issue (I do not see the relevance of it to New Insights in Veterinary Endoscopy: From Diagnosis to Therapy – so, I suggest directing it to another section in animals.
Author Response
We express our sincere gratitude for your valuable insights and thorough examination of our work. Your thoughtful suggestions have greatly contributed to the refinement of our study.
We acknowledge the limitations inherent in the retrospective nature of our research, particularly in the review of electronic medical records. This approach constrained our ability to differentiate between various forms of chronic enteropathy as defined by current classifications, such as FRE, ARE, IRE, and NRE.
The primary objective of our study was to assess the potential utility of ongoing hematological ratios in canine IBD. The identification of alterations in certain ratios (NLR, MLR, and PLR) among IBD patients compared to healthy subjects serves as a foundation for our ongoing efforts. We are committed to conducting future prospective studies that will delve deeper into these findings.
In response to your suggestion regarding a Special Issue, we plan to approach the Publisher to inquire about the possibility of including our article, should it be accepted, in the upcoming Special Issue titled "Diseases of the small and large intestines, liver, and pancreas in small animals."
Once again, we appreciate your time, expertise, and commitment to advancing the quality of scientific research. Your feedback has been invaluable, and we look forward to the possibility of contributing further to the understanding of canine IBD.